# Site-dependent reactivity of MoS$_2$ nanoparticles in hydrodesulfurization of thiophene

Norberto Salazar [1,3,5], Srinivas Rangarajan [2,4,5], Jonathan Rodríguez-Fernández [1], Manos Mavrikakis [2✉] & Jeppe V. Lauritsen [1✉]

The catalytically active site for the removal of S from organosulfur compounds in catalytic hydrodesulfurization has been attributed to a generic site at an S-vacancy on the edge of MoS$_2$ particles. However, steric constraints in adsorption and variations in S-coordination means that not all S-vacancy sites should be considered equally active. Here, we use a combination of atom-resolved scanning probe microscopy and density functional theory to reveal how the generation of S-vacancies within MoS$_2$ nanoparticles and the subsequent adsorption of thiophene (C$_4$H$_4$S) depends strongly on the location on the edge of MoS$_2$. Thiophene adsorbs directly at open corner vacancy sites, however, we find that its adsorption at S-vacancy sites away from the MoS$_2$ particle corners leads to an activated and concerted displacement of neighboring edge S. This mechanism allows the reactant to self-generate a double CUS site that reduces steric effects in more constrained sites along the edge.

[1] Interdisciplinary Nanoscience Center (iNANO), Aarhus University, Gustav Wieds Vej 14, DK-8000 Aarhus C, Denmark. [2] Department of Chemical & Biological Engineering University of Wisconsin–Madison, Madison, WI 53706, USA. [3] Present address: Centro de Investigación en Tecnologías de la Información y las Comunicaciones (CITIC-UGR), University of Granada, 18014 Granada, Spain. [4] Present address: Department of Chemical and Biomolecular Engineering, Lehigh University, Bethlehem, PA 18015, USA. [5] These authors contributed equally: Norberto Salazar, Srinivas Rangarajan. ✉email: emavrikakis@wisc.edu; jvang@inano.au.dk

C atalytic hydrodesulfurization (HDS) is an industrial process applied to remove sulfur heteroatoms from gas oil feeds to reduce sulfur content in fuels and petrochemicals to a <10 ppm level, compatible with legislative specifications. The primary catalysts used for removal of sulfur from crude oil are based on molybdenum disulfide ($MoS_2$) nanoparticles, often promoted with Co or Ni, and supported on an alumina substrate[1,2]. Extensive studies involving reaction kinetics, atom-resolved microscopy, X-ray and IR spectroscopy, and density functional theory (DFT) have made it evident that the active sites are located on the edge of single-layer $MoS_2$ catalyst[3–10]. The nature of these sites on the catalytically active edges was studied by selectively adsorbing probe molecules such as CO or NO[11–14]. These experiments allowed the titration of the number of these sites and its correlation with catalytic activity. Thiophene ($C_4H_4S$) has furthermore often been used as an S-containing reactant molecule to probe HDS activity of a catalyst[15–20]. This simple five-membered aromatic heterocyclic molecule is relatively inert as its aromatic ring makes its C–S bond quite resistant to rupture. Despite the many fundamental studies devoted to a theory- and experiment-based understanding of the HDS of thiophene a comprehensive mechanistic understanding of this process is still lacking. HDS may proceed by two independent routes: the dominant route for thiophene is the direct desulfurization (DDS) pathway, which involves the direct hydrogenolysis of the C–S bond. The generic type of active site considered for the DDS pathway is a coordinately undersaturated site (CUS) located at a sulfur vacancy ($V_S$) on an edge site formed by reaction with $H_2$[2,19–27]. The concentration of the CUS sites has been shown to be dependent on the chemical potential of hydrogen and sulfur in the gas phase and the type of $MoS_2$-edge[5,28–35]. In parallel to DDS, the hydrogenation pathway (HYD) can proceed, in which thiophene adsorption is followed by hydrogenation before C–S bond scission[36–40]. The HYD route involves a different type of active site[4], proposed to be brim sites located at the top of the $MoS_2$ particle edges[6,8]. The HYD route becomes more pronounced compared to DDS for large S-containing molecules such as dibenzothiophene and prevailing for alkyl-substituted derivatives, such as 2,5-dimethyldibenzothiophene[41,42]. However, whether the DDS route can still proceed efficiently for such large molecules, which are sterically hindered in their adsorption on a single $V_S$ site, remains an open question and a target of intense industrial HDS catalyst development due to the comparatively lower hydrogen consumption in DDS.

Herein, we use a combination of atom-resolved scanning tunneling microscopy (STM) and DFT on corresponding $MoS_2$ nanoparticle structures to determine the precise adsorption configuration of thiophene on all types of CUS sites present on the $MoS_2$-edges and corner sites present under HDS conditions. We first investigate the formation of $V_S$ sites by monitoring the distribution of sulfur vacancies and their associated formation energies. We then expose $MoS_2$ to thiophene, which allows quantification of the accessibility of $V_S$ sites grouped by their position on the $MoS_2$-edges and the corresponding adsorption configuration of thiophene. Thiophene adsorption is possible on all observed $V_S$ sites. However, whereas thiophene adsorbs directly in open $V_S$ sites at the corners between edges, the adsorption in $V_S$ sites located at the interior of the edge is observed to occur via a displacement of a neighboring S atom leading to simultaneous formation of an $S_2$ dimer and thiophene adsorbed on a double-$V_S$ CUS site. The main finding is therefore that the adsorption of an S-containing molecule may generate its own, more accessible adsorption site, which facilitates the subsequent desulfurization of the molecule. This mechanism contrasts prevalent conclusions on the HDS active site as a static vacancy configuration, and can thus explain the apparent DDS reactivity observed for larger molecules than thiophene, such as dibenzothiophene; it further opens up the discussion for a more accurate description of catalytic inhibition in co-processing of e.g. aromatics and O- and N-bearing reactants in hydrotreating[25,43–46].

## Results

**Site-dependent sulfur vacancy formation.** The reactivity of thiophene on $MoS_2$ was first experimentally evaluated by atom-resolved STM imaging of single-layer $MoS_2$ nanoparticles. We have specifically characterized S vacancies on hydrogen-activated $MoS_2$-edge structures and then assessed the adsorption of thiophene on these sites by use of a model system composed of well-defined $MoS_2$ nanoparticles synthesized on an Au(111) single crystal surface (see methods section). Figure 1 illustrates an atom-resolved STM image of a $MoS_2$ nanoparticle (Fig. 1a) together with a top view of a structural model (Fig. 1b) based on previous extensive atom-resolved characterization and theoretical modeling[27,47–49]. The $MoS_2$ particle in Fig. 1a represents the HDS active state (denoted by $r$-$MoS_2$) induced in the experiment by dosing $H_2$ gas at elevated pressure and temperature as presented in ref. [50] (see "Methods" section). The resulting $r$-$MoS_2$ particle morphology is represented by a truncated triangular shape, which is bounded by two different types of edges referred to as Mo- and S-edges, respectively. The hydrogen activation leads to a reduced overall 50% S-coverage on the Mo-edge, represented by S monomers located at a bridge position between edge Mo atoms[50] (Fig. 1b). The same 50% S-coverage on the Mo-edge structure was concluded to be present in situ at 1 bar $H_2$ pressure in a recent reactor STM study[51]. Importantly, in Fig. 1a, a number of individual S vacancies can be clearly identified along the Mo-edge imaged as sites with an extinguished intensity in the STM image (see red arrows)[50]. Atomic defects are sometimes present within the basal plane too (e.g. in Fig. 1a), but these are not induced by the hydrogen treatment and their location suggest an impurity atom on the metal lattice of $MoS_2$ rather than a basal plane $V_S$. Based on a statistical analysis, the total number of vacancies at the Mo-edges is modest overall, corresponding to an average probability of 16% for a vacancy being formed at any edge site. This observation is, however, in full accordance with previous theoretical modeling of the S-vacancy formation, which results in a rather large energy cost required for $V_S$ formation by removal of S atoms from the 50% S covered Mo-edge[34,52].

Importantly, our analysis shows that the specific $V_S$ probability varies significantly among the different sites. To see this, we labeled S atomic positions with a letter in the ball model representations (Fig. 1b, c) to identify S vacancies on the corner (C), adjacent corner (A) and in the middle (M) positions. These sites define the possible edge positions on the Mo-edge lengths most abundantly present in the synthesized $MoS_2$ nanoparticle ensemble (Supplementary Fig. 1), corresponding to 4-6S monomers on the edge as shown in the side-view ball models in Fig. 1c. A breakdown of our statistical materials from the experimental images into each edge length (Supplementary Fig. 2) shows only a very slight variation of the vacancy numbers with the edge length, so for the experimental analysis we consider C, A and M sites on differently sized edges to be similar in terms their vacancy formation probability. The example in the STM image in Fig. 1a already points to the preferential formation of $V_S$ mainly at A and M locations (see top-view ball model). Figure 2a illustrates the frequency of observing a $V_S$ on these specific sites obtained by a direct counting of missing S atoms in all our STM images. Here, the most probable S vacancies are located at the A positions, with a $V_S$ probability at that site being 0.25 (i.e. 1 out of 4 of all A sites counted in the STM images were observed

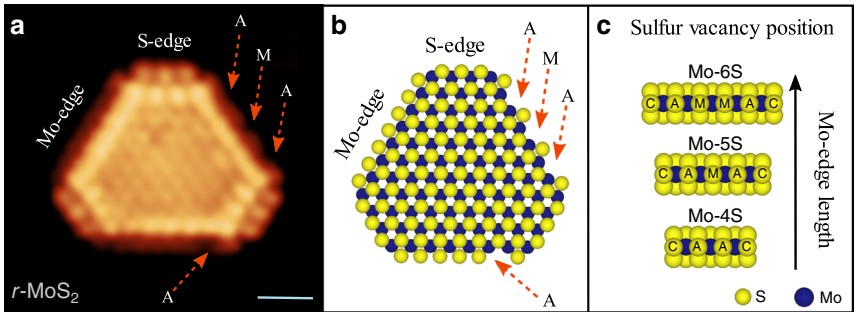

**Fig. 1 Atomic-scale structure of MoS$_2$ nanoparticle with sulfur vacancies formed on the edges. a** Atom-resolved STM image recorded at room temperature of a single-layer MoS$_2$ nanoparticle exposed to hydro-reductive conditions (r-MoS$_2$) at 673 K and $10^{-4}$ mbar of H$_2$. The red arrows indicate the position of individual sulfur vacancies (V$_S$) which in this case include 3 V$_S$ on A position and 1 V$_S$ on an M position. STM parameters: $V_t = -0.32$ V, $I_t = -0.42$ nA. Scale bar is 1 nm. The STM image in **a** of r-MoS$_2$ is adapted from ref. [50] (reprinted with permission). **b** Left part: Top-view ball model of the r-MoS$_2$ particle in (a) (S: yellow, Mo: blue). The truncated MoS$_2$ crystal is bounded by three short S-edges and three longer Mo-edges with a nominal 50% S-coverage consisting of S-monomers at a bridge position between Mo atoms, as in ref. [50]. Sulfur vacancies are indicated by a red arrow. **c** Side-view ball model of the Mo-edge with a nominal coverage of 50% for three different edge lengths (4, 5, and 6 monomers, respectively). The overlaid notations for edge site locations are (C) corner, (A) adjacent to corner and (M) middle positions, respectively.

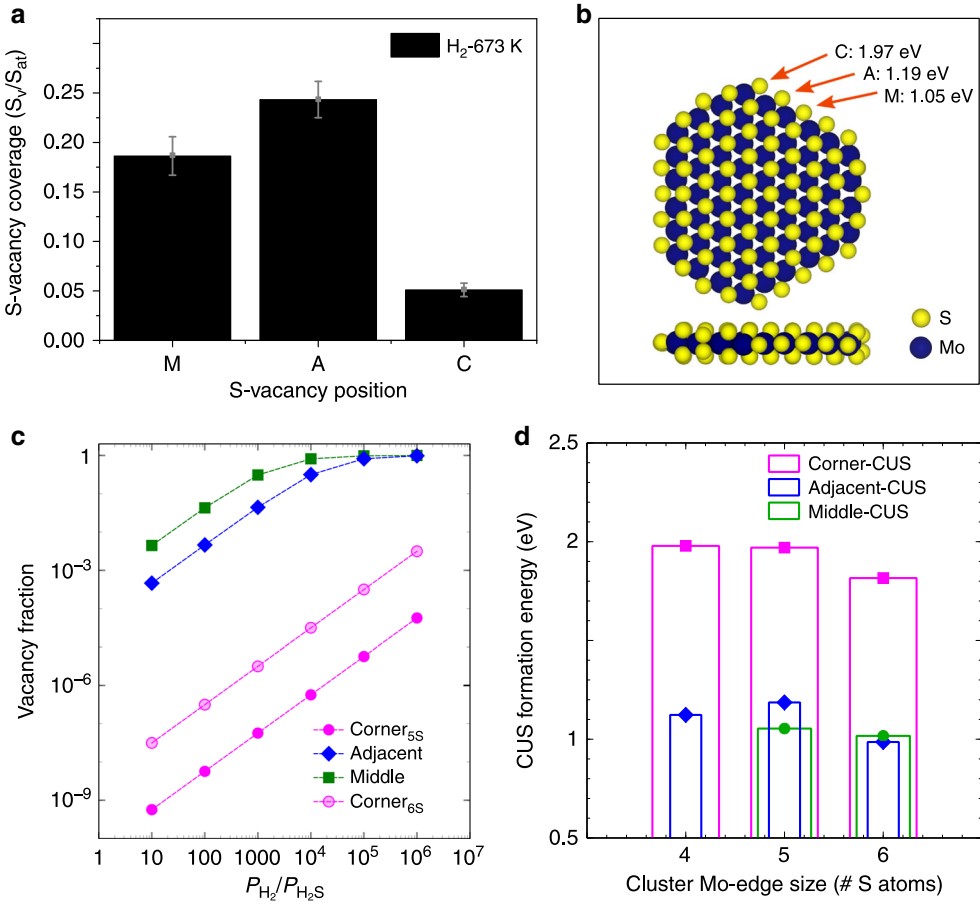

**Fig. 2 Site-dependent vacancy formation on MoS$_2$-edges. a** Bar plot showing the observed frequency of finding a V$_S$ on corner (C), adjacent (A) and middle (M) sites as determined from atom-resolved STM images. **b**. Top and side-view ball model of the r-MoS$_2$ nanoparticle (5S monomer long Mo-edge) used in the DFT model. The calculated V$_S$ formation energies ($E_S$) for the M, A, and C edge sites, respectively, are given in eV (positive values are endothermic, see supplementary information). **c** Theoretically predicted V$_S$ fraction (log scale) at each site as a function of the ratio of H$_2$ and H$_2$S partial pressures at 673 K. **d** Size-dependent variation of the V$_S$ formation energy (CUS formation) for particle models exposing 4 to 6S monomers along the Mo-edge (corresponding structures are shown in Supplementary Fig. 4). Corner position (square), Adjacent position (diamond), Middle position (circle).

to have a V$_S$). It is slightly less probable to find S vacancies at M positions, with 0.18 of all M sites containing a V$_S$. The number of S vacancies observed directly at the corner site between the Mo-edge and S-edge is considerably lower at 0.06,

indicating a higher initial stability of the terminal S monomer at the corner site.

DFT calculations were performed to provide additional information on the probability of V$_S$ formation. The model

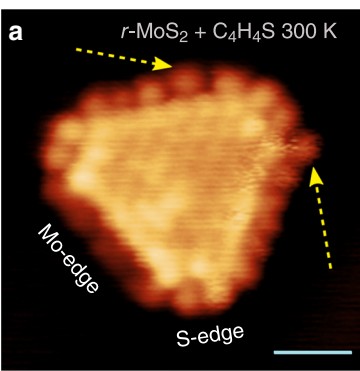

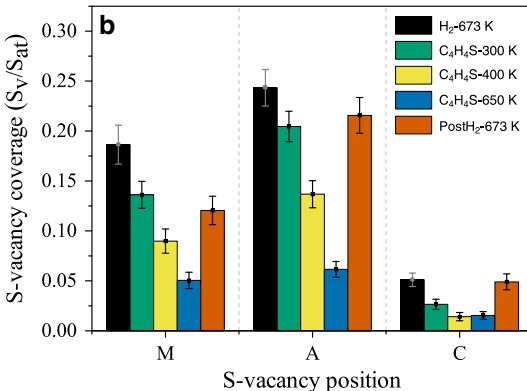

**Fig. 3 Thiophene adsorption on MoS$_2$-edges. a** Atom-resolved STM image of a $r$-MoS$_2$ nanoparticle (Mo-6S edges), which was subsequently reacted with thiophene gas at room temperature (300 K). The superimposed arrows indicate the locations attributed to adsorbed thiophene. STM parameters: $V_t$= −0.22 V, $I_t$=−0.40 nA. Scale bar is 1 nm. **b** Statistical plot showing the variation of the $V_S$ fraction for each edge site position (M, A, and C) obtained by counting S vacancies in atom-resolved STM images, compiled from images with 4,5 and 6S atom positions on the Mo-edge. The data reflect a temperature series for the H$_2$ activated sample (black bins), and where thiophene is dosed at 300 K (green), 400 K (yellow), and 650 K (blue), respectively. The orange bins show the restored $V_S$ vacancy distribution after post-reacting the thiophene-exposed sample in H$_2$ at 673 K. A statistical breakdown for each edge length is provided in Supplementary Fig. 2. Error bars reflect the statistical uncertainty based on the standard error.

chosen to represent experimentally observed MoS$_2$ nanoparticles was a freely standing truncated MoS$_2$ nanoparticle exposing a short S-edge with 100% S and a five S-monomers long Mo-edge (Mo-5S) comprising of sites of the C, A, and M type (Fig. 2b). The MoS$_2$ nanoparticle models are important in our study since they possess higher site heterogeneity and allow a greater degree of S mobility on the edge, as opposed to semi-infinite stripe models used in previous work which only contain M sites[5,18,19,24]. We did not specifically include the Au substrate here, as recent calculations involving semi-infinite periodic calculations indicated that the substrate has a negligible effect on the bonding of S monomers to the Mo-edge[50]. As listed in Fig. 2b, the energy required to form S-vacancies ($E_S$) (see "DFT calculations" in Methods) on the C, A, and M locations of the MoS$_2$ nanoparticle is endothermic in all cases. However, the vacancy formation on the corner C site is significantly more unfavorable ($E_S = 1.97$ eV) than the A or M position ($E_S = 1.19$ eV and 1.05 eV, respectively). For reference, our corresponding calculation of $E_S$ on a semi-infinite stripe model gave a value of 0.99 eV, which is slightly lower than that of the M position. To quantitatively account for the effect of temperature and partial pressures of H$_2$ and H$_2$S in the experiment, we calculated the Gibbs free energy of $V_S$ formation from which the vacancy fraction at each of the three positions considered can be calculated (see "Theory methods"). We note that some H$_2$S is present in the background gas even if pure H$_2$ is used, due to residual gas and the fact that H$_2$S is released during S vacancy formation, corresponding to a H$_2$/H$_2$S ratio of approximately $10^3$–$10^4$ or higher (estimated from gas analysis using mass spectroscopy). Figure 2c shows the calculated vacancy fraction on the three locations at the experimental temperature of 673 K and a total pressure of $10^{-4}$ mbar as a function of the relative partial pressure of H$_2$ and H$_2$S from sulfur rich (H$_2$/H$_2$S > 1) to sulfur poor conditions (H$_2$/H$_2$S > $10^7$). The model predicts a significant population of S vacancies at the M and A positions in the range of H$_2$/H$_2$S ratio between $10^3$ and $10^4$, in agreement with our experimental findings (Fig. 2a). Going to more sulfur poor conditions (H$_2$/H$_2$S > $10^5$) from this point leads to almost unity vacancy fraction reflecting fully stripped M and A sites equivalent to a naked Mo-edge, which we never observed experimentally. On the other hand, the model predicts a very small fraction of corner (C) vacancies, below $10^{-7}$ as opposed to ~6% found in experiments. This difference may be

due to the intrinsic errors in DFT (of 0.1–0.2 eV), which can introduce uncertainties in the calculated vacancy population. Furthermore, while the Au substrate is expected to not significantly influence $E_S$ on the M and A sites, which is captured well in semi-infinite models[50], its influence on the corner site could be a source of error in our calculation of vacancy coverage for C positions in Fig. 2c. The DFT modeling also predicts that the $V_S$ formation energies on C sites are dependent on the edge length. Figure 2d shows that the sulfur vacancy formation energy ($E_S$) at the corner site (see also Supplementary Fig. 4) decreases from ~2 eV to 1.8 eV (see Supplementary Fig. 4) upon going from Mo-edge length of four to six. Hence, this dependency will contribute to an increase of the overall $V_S$ fraction compared with the 5 S-atom Mo-edge, as shown in Fig. 2c (lightly shaded pink circles). The increasing trend for the $V_S$ probability for the corner sites is also noted in the breakdown of the experimental data for different edge lengths in Supplementary Fig. 2. For comparison, the calculated $E_S$ values at the M site in Fig. 2d (only present in 5S and 6S Mo-edges) shows that the vacancy coverage should be size independent, which is consistent with our STM observations (Supplementary Fig. 2). Furthermore, for the A sites, DFT predicts a variation where the S atom is hardest to remove on the 5S Mo-edge (1.19 eV for 5S compared to 1.12 eV and 0.99 eV for 4S and 6S respectively in Fig. 2d). We note that the DFT functional tends to trimerize Mo atoms particularly on the 5S Mo-edge which makes it generally easier to remove the S atom on the longer-bridge (e.g. M site of 5S) than others. We posit that this effect is the reason for variation of $E_S$ values with Mo-edge length, which is not reflecting the variation in the experiment.

**Site-dependent thiophene desulfurization reactivity.** To probe the specific affinity of different types of $V_S$ sites towards the adsorption of S-containing molecules, we then exposed the $r$-MoS$_2$ nanoparticles to thiophene vapor at different reaction temperatures, ranging from room temperature to 650 K. By means of STM, we then imaged the structural modifications resulting from adsorption of thiophene on each edge site. Figure 3a shows an atom-resolved $r$-MoS$_2$ nanoparticle exposed to thiophene vapor ($10^{-6}$ mbar) at room temperature (300 K). After exposure to thiophene, the $r$-MoS$_2$ nanoparticles do not reveal any interaction with thiophene on the basal plane, in accordance

with the expected chemical inertness of the $MoS_2(0001)$ basal plane. On the longer Mo-edge, however, we observe a changed structural pattern indicative of thiophene adsorption, where the S in thiophene is expected to adsorb at the sulfur vacancy. For example, the edge protrusions indicated by yellow arrows at a C and A site on the $MoS_2$ nanoparticles in Fig. 3a suggest thiophene being adsorbed at these sites. However, since individual thiophene molecules in the STM image is a protrusion with an apparent size only slightly larger than that of individual S atoms, a definitive distinction of individual thiophene molecules from terminal S atoms at the Mo-edges is challenging. The corresponding short S-edges did not show changes in the edge structure upon thiophene exposure (see Supplementary Fig. 3), which is indicative of thiophene not adsorbing on the 100% S coverage S-edge. To evaluate the site-dependent reaction in a quantitative way, we instead use a statistical method based on atom-resolved images that counts the number of S vacancies that have become filled at C, A, and M positions after exposure to thiophene as a function of temperature. The comparison of the $V_S$ observations before (black bars) and after dosing thiophene molecules at $10^{-6}$ mbar at 300 K, 400 K and 650 K substrate temperatures, respectively, is depicted in Fig. 3b. By exposing thiophene at 300 K, clear variations in the S-vacancy coverage are observed (green bars). The value at position A decreases from 0.25 to 0.20, reflecting that thiophene reacts here, whereas the fraction of S vacancies at position M does apparently not change within the statistical error. In the case of the corner position C, the $V_S$ fraction drops from 0.06 to less than 0.03. These changes demonstrate that thiophene molecules can fill the C and A vacancy sites at the Mo-edges already at room temperature without a substantial activation barrier. Dosing thiophene at 400 K shows a further decrease in the vacancy fraction for all the positions (yellow bars). At this temperature, the corner sites seem to be almost fully occupied by thiophene molecules. The vacancy fraction at A still decreases more rapidly than at position M, suggesting that the inhibition of thiophene adsorption at position M may be related to larger steric effects of thiophene adsorption at these sites. Experiments carried out at even higher temperatures (650 K, blue bars in Fig. 3b) show that adsorption on the M sites is now enabled, and the overall $V_S$ fraction now decreases to 0.05, 0.06, and 0.02 for positions M, A, and C, respectively. The onset of adsorption on A and M sites at higher temperature, compared with C, reflects a higher intrinsic kinetic barrier for thiophene adsorption on these sites. An energy barrier for adsorption is expected when the access is sterically limited and thermal activation may thus be required when relaxation at the adsorption site is part of the adsorption mechanism. A final treatment in reductive $H_2$ conditions was performed to evaluate for irreversible changes due to thiophene adsorption, using the same $H_2$ dosing parameters as in the initial preparation of r-$MoS_2$. The $H_2$ exposure is seen to remove features associated with adsorbed thiophene and restores the $V_S$ distribution (orange bars) among the three site types close to the initial values. Since high temperature alone is not enough to desorb the thiophene (Fig. 3b, blue bars), we consider that the process is likely due to a hydrogenolysis reaction that produces an organic product and $H_2S$, thereby completing a full catalytic cycle. This shows that thiophene adsorption and regeneration of $V_S$ can be considered to be reversible at the investigated temperatures.

## Thiophene adsorption modes on $MoS_2$-edge and corner sites.

To address differences in the bonding affinity of thiophene at C, A, and M-type $V_S$ sites, we performed plane wave DFT calculations for thiophene adsorption on pre-formed S-vacancies along the edge of the $MoS_2$ nanoparticle model (see Fig. 2b) and identified the most energetically stable adsorbed states (see Fig. 4

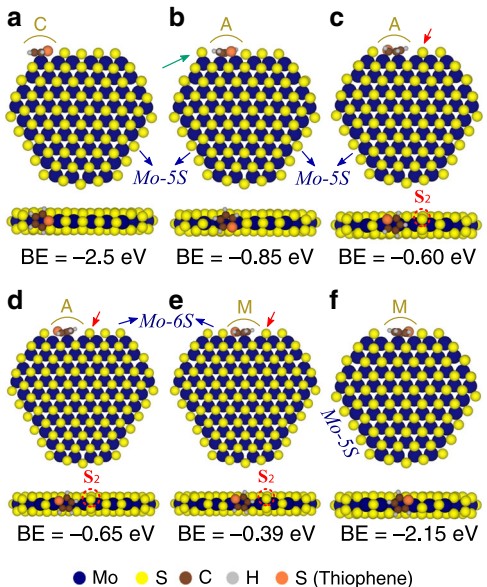

**Fig. 4 Thiophene adsorption modes from theory.** Top and edge views of calculated minimum energy structures for the adsorption of thiophene on **a** the corner vacancy, **b** adjacent vacancy with S atom on the C location shifting from the bridge to top position, **c** adjacent vacancy with S atom on the M location dimerizing with the neighboring S atom, **d** adjacent vacancy similar to **c** but on a longer (six-atoms-long) edge, **e** middle (M) vacancy with the S atom on the second M location dimerizing with the neighboring S atom on a six-atom-long edge, and **f** contiguous vacancies on adjacent and middle location. The $MoS_2$ nanoparticle show different adsorption configurations with their respective adsorption energy. The computed binding energies (BE) are with respect to the corresponding non-rearranged nanoparticle without the adsorbate (and thiophene in the gas phase). That is, the reference particle for **a** is one with a vacancy at the corner, for **b–d** is one with a vacancy at the A site, for **e** is one with a vacancy at the M site and **f** is one with vacancies at both A and M sites. Red arrows and dashed circles indicate the formation of S-dimers ($S_2$). The green arrow points to the displaced S atom on the corner. Arrows and labels in blue indicate the edge length on the Mo-edge used in the calculation.

for most stable structures and Supplementary Table 1 for alternative structures). We then compare these states with the corresponding edge structure observed on these sites in atom-resolved STM images, obtained after thiophene exposure at room temperature (Fig. 5). The most stable adsorbed thiophene state corresponds to adsorption directly at the corner S-vacancy site C (Fig. 4a). In this state, thiophene binds strongly (binding energy, BE = −2.50 eV) to the exposed Mo atom available at the C site. This is in line with our experiments (Fig. 3b), showing that the C vacancy sites react with thiophene readily at room temperature. In the STM images shown in Fig. 5a, we see a bright (see also linescan I in Fig. 5d), larger, and slightly displaced protrusion at the corner site (indicated with a C in the superimposed model structure), consistent with the presence of a thiophene molecule adsorbed in configuration shown in Fig. 4a.

The most stable adsorption mode found for thiophene on the A vacancy site is shown in Fig. 4b. In this state, thiophene adsorption is still exothermic by ~ −0.85 eV, but much less favorable than for corner S-vacancies, indicating that steric hindrance is significant on the A site. Importantly, the DFT calculation shows that the adsorption introduces a significant structural modification to "make space" for thiophene, thus displacing the neighboring S atom in the r-$MoS_2$. Specifically, as the thiophene is adsorbed onto a S-vacancy on the A site, it either pushes away a neighboring S atom on the corner from a bridge to

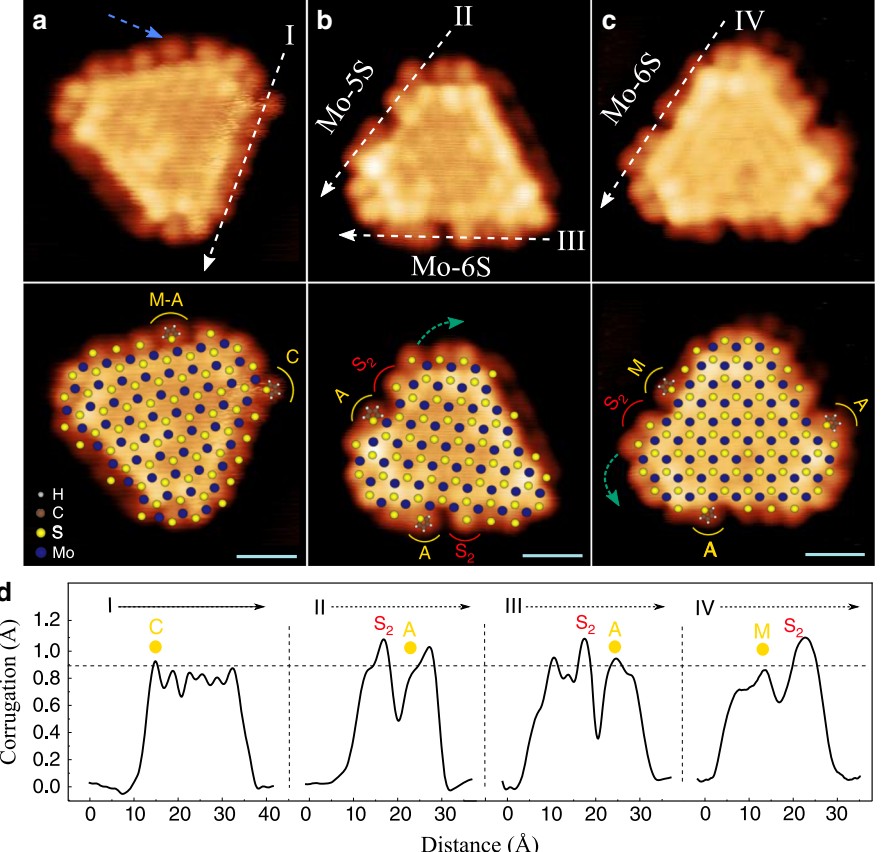

**Fig. 5 Thiophene adsorption modes in STM. a–c** Three cases of STM images of $r$-MoS$_2$ nanoparticles after exposure to thiophene at 300 K shown with a superimposed model shown below each image. The white dashed lines correspond to the direction of the edge profile shown below. The blue dashed arrow indicates thiophene adsorption on M and A locations. The STM images with the superimposed nanoparticle lattice show the location of the thiophene molecules (yellow marks) and sulfur dimer formation (red marks), and atomic displacement direction indicated with green dashed arrows. STM parameters for **a** $V_t = -0.22$ V, $I_t = -0.40$ nA; **b** $V_t = -0.25$ V, $I_t = -0.33$ nA; and **c** $V_t = -0.15$ V, $I_t = -0.28$ nA. Scale bar is 1 nm in all images. **d** Linescans of the Mo-edge I, II, II, and IV represented in the STM images.

a top position (see the green arrow in Fig. 4b) or displaces a neighboring S atom on the interior (Fig. 4c, d) to its neighboring site leading to a relatively less exothermic adsorption (BE = −0.60/−0.65 eV for Mo-5S and Mo-6S edges, respectively). Indeed, this latter S displacement leads to the formation of a stable S$_2$ dimer (formally $S_2^{2-}$) (see red arrow and dashed circle in Fig. 4c) and places the thiophene in coordination with a double $V_S$ vacancy located over both the A and M sites. Note in Fig. 4c that the S in thiophene coordinates to one of the under-coordinated Mo atoms, whereas that carbon ring system is placed over the Mo which has become accessible due to the displacement of the S. Our calculations identified additional favorable thiophene adsorption states with the concurrent formation of S dimers and corner S atop a corner Mo atom (with BE ~ −0.5 to −0.6 eV, see structures (i)–(iii) of Supplementary Table 1). Our calculations also showed that thiophene adsorption on a S-vacancy on the M site can generate a favorable adsorption site by displacing all S atoms on the edge to a top position without S$_2$ formation (see structure (iv) of Supplementary Table 1); we expect, however, that such a concerted displacement may involve a large kinetic barrier due to an additive contribution of each S displacement. As shown in Fig. 4e, for adsorption at the M vacancy site, the S$_2$ dimer can also be formed on the other side of the edge closer to the C site, in a configuration with an exothermic BE of −0.39 eV (note here we used a six S-atom-long Mo-edge (Mo-6S)).

The STM images in Fig. 5b, c provide the experimental evidence that adsorption on A and M sites occurs via S displacement. To observe this, we point to the contrast asymmetry around the distinct dark edge region denoted with an A and S$_2$ or M and S$_2$, respectively, in the lower part of Fig. 5b, c (see also linescans II, III and IV in Fig. 5d). Here, the S$_2$ dimer is located at the high contrast protrusion on only one side and the dark spot reflects the rather open region between thiophene and the S$_2$ dimer seen in all cases in the structures in Fig. 4c–e. The edges reflecting linescans II and IV correspond to the adsorption of thiophene where the S$_2$ dimer is located either away from the nearest corner site or towards the corner position. We also observe the modeled displacement of the corner S monomer from a bridge to on top in configuration 4b, as indicated by the position of the protrusion at the green arrow in the superimposed image below Fig. 5b.

Our calculations show that S$_2$ dimerization, per se, is typically endothermic with energies ranging from 0.7–1.9 eV depending on the location of the dimers and the concomitant vacancy (see Supplementary Table 2). Since the adsorption with dimerization is, in all cases, exothermic, we checked if thiophene adsorption and S displacement occurs with a concerted transition state or sequentially. As shown in Supplementary Table 3, the barrier for S-dimerization starting from an edge vacancy and creating two adjacent vacancies without thiophene is very high (~2.1 eV). The barrier for the concerted step is 0.8 eV lower than the sequential

pathway (at ~1.3 eV), which is fully consistent with the STM observation in Fig. 3b that thiophene adsorption on A and M sites is favorable, but with a surmountable reaction barrier.

Finally, we note that we occasionally can observe thiophene adsorption on M and A locations without the characteristic formation of the $S_2$ dimer in STM images (blue arrow in Fig. 5a), which could indicate thiophene adsorption directly in a double $V_S$. The direct adsorption of thiophene onto a pre-existing double $V_S$ site is highly favorable with BE energy of $-2.16$ eV (Fig. 4f and structures (v)–(vi) of Supplementary Table 1). We, however, never observe two adjacent vacancies experimentally. This is in accord with our calculated formation energy of 3.4 eV for the double $V_S$, leading to two S vacancies in adjacent M and A positions, which is greater than the sum of the individual $V_S$ formation energies at those positions. Instead, we frequently see two vacancies in the next-nearest neighboring positions (Fig. 1a, b). Starting from such a configuration wherein two vacancies are separated by a single S monomer, our calculations determine that the sulfur migration to a neighboring vacancy and the concurrent adsorption of thiophene on the newly created adjacent vacancies is energetically favorable by $-1.1$ eV (structure (vii) in Supplementary Table 1). This serves as an additional evidence that thiophene adsorption displaces edge S atoms in a concerted manner to create enough room at the adsorption site.

In the HDS catalytic cycle, the overall DDS reaction on a sulfur vacancy site ($V_S$) is considered to take place by thiophene adsorption, subsequent S hydrogenolysis, followed by $V_S$ regeneration and $H_2S$ release. A favorable reaction site should thus exhibit an appropriate balance between a favorable adsorption energy of thiophene (BE) and a reasonable $V_S$ formation energy ($E_S$). Our findings indicate that corner sites in a MoS₂ nanoparticle are very favorable for thiophene adsorption, but our modeling also shows that that vacancy formation is so energetically expensive that a very low fraction of the corner (C) sites will contain a $V_S$ under steady-state reaction conditions. This is in line with recent findings on corner site reactivity in Co promoted MoS₂[23,34,35]. The HDS activity in thiophene DDS is therefore mainly ascribed to Mo-edge sites. We find that $V_S$ formation is less expensive on the M and A sites of the Mo-edge, and both sites are shown to be able to adsorb thiophene. Importantly, thiophene adsorption itself leads to restructuring of the S atoms on neighboring sites around the initial $V_S$ sites and the formation of a more open site, in some cases reflecting a double $V_S$ site. This can either happen by formation of a neighboring $S_2$ dimer, if the starting point is a single $V_S$, or by merger of two next-nearest neighbor $V_S$. The observations illustrate that adsorption of S-containing molecules on the edges does not follow a simple Langmuir (checker-board) adsorption model, but should rather be considered a case where the adsorption site dynamically adjusts to the adsorbate by S displacement. We expect that these dynamic arrangements are relevant in general for HDS reactions of a wider range of compounds, as they explain why there is "enough room" for molecules such as dibenzothiophene to adsorb and undergo DDS on the edges of MoS₂. The observations reported here, therefore, have profound implications on our understanding of the reactivity of MoS₂-based hydrotreating catalysts, in particular for HDS and hydrodenitrogenation of complex refractory feedstock composed of a blend of large S and N containing compounds as well as for emerging technologies such as hydrodeoxygenation of lignocellulosic biomass.

## Methods

**Experimental details.** The experimental approach is based on the synthesis method of well-defined nanocrystals of MoS₂ on Au(111)[27,30,53]. The synthesis of MoS₂ nanoparticles on Au(111) was carried out on a clean Au(111) single crystal in a standard ultra-high vacuum (UHV) equipped with a homebuilt Aarhus-type variable temperature STM. The sample temperature was measured with a K-type thermocouple in contact with the backside of the Au crystal. MoS₂ nanoparticles were synthesized by physical vapor deposition of Mo evaporation onto the gold surface using an e-beam evaporator (Oxford Applied Research EGCO-4). $H_2S$ gas (AGA, purity 99.8%) was dosed for sulfidation corresponding to a background pressure of $1.0 \times 10^{-6}$ mbar for 15 min, while the sample temperature was 400 K. In order to obtain full crystallinity of the MoS₂ nanoparticles, the samples were post-annealed in $H_2S$ atmosphere to 673 K for 10 min. The MoS₂ nanoparticles were subsequently activated under reducing conditions by back-filling the UHV chamber with $H_2$ (99.999%, Praxair)[50]. The reduced (active) MoS₂ nanoparticles ($r$-MoS₂) used as the starting point of the experiments reported here were obtained by annealing the fully sulfided samples to 673 K in of $10^{-4}$ mbar of $H_2$. For thiophene (Sigma Aldrich, 99% purity) dosage, the liquid was kept in a glass container and admitted to the UHV chamber through a leak valve connected to a stainless steel tube directed to the sample surface. Prior to dosing, the thiophene liquid was purified by several freeze-pump-thaw cycles to remove dissolved gas. The gas purity was checked and $H_2S$ and $H_2$ partial pressures were measured by a quadrupole mass spectrometer (Hiden Analytical). The samples were exposed to thiophene at three different substrate temperatures: 300, 400, and 650 K; a background pressure of $1.0 \times 10^{-7}$ mbar was used for 5 min during the thiophene dosage. After each thiophene exposure, the sample was cooled down to room temperature and transferred to the STM for a local inspection. STM images were recorded at room temperature using etched W tips in the constant current mode. The tunneling parameters are noted in the image text as $V_t$ (tunneling bias) and $I_t$ (tunneling current).

**DFT calculations.** All calculations (unless specified otherwise) were carried out for a single-layer hexagonal nanoparticle model of MoS₂ with Mo-edge containing two to six molybdenum atoms while the S-edge contains four atoms, reflecting the typical truncated triangular shape of a single-layer particle observed in STM experiments. Two layers of sulfur atoms sandwich the molybdenum layer such that they are in the trigonal prismatic positions characteristic of the 2H phase of MoS₂. Further, most of the calculations involve a Mo-edge of six molybdenum atoms (and 5S monomers, denoted Mo-5S), also consistent with our observations that metal edges are longer than the sulfur edges. All figures in this document with MoS₂ structures and adsorption configurations show Mo atoms in blue, S atoms in yellow, C atoms in brown, and H atoms in gray. The Au(111) support was not included in these calculations; while the support can influence the stability of the edges, the single-layer stripe model used in periodic calculations have successfully captured the correct edge morphologies under sulfidation and reducing conditions, as inferred from STM studies We, therefore, suggest that the given model captures the local electronic structures adequately to provide a comparative analysis between different locations along the periphery of the MoS₂ nanoparticles used in experiments. We further assume that the sulfur edge is 100% S-decorated while the metal edge is 50% S-decorated, consistent with ab initio phase diagrams and STM observations[50]. Importantly, we introduce S-vacancies on the Mo-edge on the center (C), adjacent (A) and middle (M) positions, as needed, to elucidate the energetics of S-vacancy formation and of thiophene adsorption on these vacancies.

The calculations were carried out with VASP[54,55], a plane wave periodic DFT code. Generalized gradient approximation and projected augmented wave (PAW) potentials[56] were used with PBE exchange correlation functional[57] and the D3 Grimme dispersion correction[58]. All calculations were carried out in a box that had at least 10 Å of vacuum between two images in all directions. Spin polarization is included in all calculations: the effect of spin was found to be negligible (~0.01 eV on the vacancy formation energy). Plane wave and density wave cutoffs of 400 and 645 eV were used, respectively. A Gaussian smearing of 0.05 eV was used, and the energies were extrapolated to 0 K. Only gamma-point sampling was used in view of the large dimensions of the supercell. The convergence criterion for geometric relaxation was set to 0.02 eV/A. The energy of vacancy formation, $E_S$, is given by

$$E_S = E_{NP,with\ CUS} + E_{H_2S} - E_{NP} - E_{H_2}, \qquad (1)$$

where $E_{NP,with\ CUS}$ is the energy of the nanoparticle with coordinative unsaturation (CUS or vacancy), $E_{H_2S}$ is the energy of $H_2S$ in the gas phase, $E_{NP}$ is the energy of the nanoparticle at the equilibrium termination (50% metal edge, 100% sulfur edge), and $E_{H_2}$ is the energy of hydrogen gas. The free energy of vacancy formation, required to estimate vacancy fraction, was computed by calculating temperature-dependent enthalpy and entropy[59]. The entropy of surface S atoms was assumed to be of vibrational origin while rotational and translational components from statistical mechanics for an ideal polyatomic species were additionally included for gaseous molecules. Vibrational entropy was computed using the Harmonic approximation, which is sufficient for all systems considered here. The binding energy of thiophene, BE, is given by

$$BE = E_{NP,\ with\ CUS+Thiophene} - E_{NP,\ with\ CUS} - E_{Thiophene}, \qquad (2)$$

where $E_{NP,\ with\ CUS\ +\ Thiophene}$ is the energy of the nanoparticle with thiophene adsorbed on the vacancy (with and without sulfur rearrangements), $E_{NP,\ with\ CUS}$ is the energy of the nanoparticle with the vacancy (without rearrangements), and $E_{Thiophene}$ is the energy of gas-phase thiophene.

For transition state calculations in Supplementary Table 1, a periodic single-layer stripe model with four rows of six Mo atoms each and commensurate S atoms was created based on our previous work[10,25,59]. We start with a 50% S covered Mo-edge and create a single vacancy in the middle of the edge to represent an internal vacancy (corresponding to an M vacancy site in the main text). The supercell was periodic in one direction (along the length of the edge) where an Mo-Mo distance of 3.19 Å was set based on our optimized lattice constant; the cell had more than 15 Å of vacuum in the other directions. The k-point sampling was $1 \times 2$ (along the edge length) x1 based on the Monkhorst Pack method[60]. All other settings for these DFT calculations were the same as above. The transition states were calculated using the climbing image nudged elastic band method[61] such that the forces on the images were less than 0.1 eV/Å.

## Data availability

The data that support the findings of this study are available from the corresponding authors upon reasonable request.

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

## Acknowledgements

The computational work was supported by the U.S. Department of Energy (DOE)-Basic Energy Sciences (BES), Office of Chemical Sciences, Catalysis Science Program, under Grant No. DE-FG02-05ER15731. Part of the calculations were conducted using supercomputing resources from the National Energy Research Scientific Computing Center (NERSC) and the Center for Nanoscale Materials (CNM) at Argonne National Laboratory (ANL). CNM and NERSC are supported by the U.S. Department of Energy, Office of Science, under contracts DE-AC02-06CH11357 and DE-AC02-05CH11231, respectively. S.R. acknowledges the use of Extreme Science and Engineering Discovery Environment (XSEDE) resources, which is supported by National Science Foundation grant number ACI-1548562. J.V.L., J.R.F., and N.S. acknowledge financial support from the Danish Research Council—Technology and Production (HYDECAT), Villumfonden and Haldor Topsøe A/S. J.V.L. acknowledges the Aarhus University Centre for Integrated Materials Research (iMAT).

## Author contributions

N.S. and J.R.F. performed the experiments. N.S. analyzed the experimental data. S.R. performed the theory. J.V.L. and M.M. planned and organized the studies. N.S. wrote the first draft. J.V.L. wrote the final version. All authors contributed to the final version of the manuscript.

## Competing interests

The authors declare no competing interests.
