## [Peer Review File · Nature Communications]

REVIEWER COMMENTS

Reviewer #1 (Remarks to the Author):

In this paper, the authors investigate a very important catalytic reaction, hydrodesulfurization. Although a lot has been found about this process, still many details remain unclear. This paper sheds new light on HDS of thiophene, and especially on the question how it adsorbs to the catalyst to have its S atom abstracted. The insight that S atoms of the catalyst move to create space for the bulky thiophene is very interesting for the community. Therefore, I advise this paper to be published, after the following minor issues have been solved.

line 67: what are exact MoS₂ structures? Please rephrase.

Fig. 1a: It is clear from the STM image that the number of vacancies is different on the three Mo edges. Is this a general observation for all MoS₂ nanoparticles? Has this been analyzed for many nanoparticles and can the difference be explained by statistical fluctuations? Or is there possibly another explanation? Please clarify this in the text.

Fig. 1c: in the bottom structure, the vacancies are named with M, but they are also adjacent to a corner, so A. Why do you choose for M, and not A? Is it because they behave more as M? Please clarify.

Fig. 2d: label x-axis. For readers less familiar with HDS and its conventions it would be more clear to call it the Mo edge instead of M-edge.

Fig. 3a: add labels to the edges in the STM image.

around line 221: if the adsorption of thiophene to certain sites is hindered by its structure, i.e. is a steric effect, how can it then be explained that the adsorption changes at higher temperature? Does the size of the vacancy change at higher T? The size of the thiophene will not change. Please clarify.

line 231: how do you know that this is due to hydrogenolysis and not just desorption of thiophene because of the higher T?

Fig. 4: please label the edges.

line 275: how does the thiophene adsorb over both the A and M sites? Probably not via the S atom to the vacancy. Please explain.

Fig. 5: this figure can be improved in the following way:

- a) remove more black so that MoS₂ can be bigger
- b) label S₂ is unreadable
- c) mention in the caption what the blue arrow in a corresponds to
- d) mention in the caption the tunneling parameters

line 335: it is not clear to me how the authors conclude on the amount of V_s under reaction conditions since their work is performed under UHV conditions. If, however, this is my mistake, and their work has been performed under reaction conditions, this should be made clearer from the text.

Reviewer #2 (Remarks to the Author):

Salazar et al. report a model study of the industrially-relevant hydrodesulfurization on MoS₂ catalysts. A model catalyst, single-layer MoS₂ nano-islands grown on Au(111), and a model reaction, thiophene (hydro)desulfurization, were employed and studied by experimental (mainly STM) and theoretical (DFT) methods.

Distinct differences in the thiophene-catalysts interactions at preformed S-vacancies at edge and corner sites (and even differently within edges) were observed. Thiophene adsorption alone leads to restructuring of the S atoms on neighboring sites around the initial vacancy sites. One could question whether the model compound thiophene really mimics the much more stable sulfur-containing compounds in gas oil feeds, but that's true for all model studies.

The work was performed on the highest technical level and all analysis and interpretations are thorough and sound. My main concern is rather that the very same model system was studied for about two decades and the major breakthroughs of this model system have already been published in general high impact journals, whereas the further progress – although important – seems rather incremental and only relevant for specialist in the field.

Reviewer #3 (Remarks to the Author):

The authors present a combined STM-DFT study of thiophene absorption on edge sites of MoS₂ nanoparticles and suggest a mechanism of S-displacement to allow incorporation of larger functional groups onto neighbouring edge sites.

In general I find that the subject of the paper is interesting, and the approach of combining STM with DFT studies is useful. However a problem with such an approach is that the images can be easily interpreted many different ways, and attention focussed just on specific aspects of the images. That appears to be the case with the current paper where the argument is heavily "driven" in favour of the model proposed by the authors, and alternative explanations and mechanisms are not covered and quantitatively dismissed sufficiently.

As some specific examples, the authors state there is no change in basal plane properties after exposure to thiophene, but do not provide any quantitative support for this statement. Basal plane vacancies appear visible in their STM images, and to justify their claim it would necessary to show for example statistics of basal plane vacancy concentration before and after thiophene treatment to demonstrate this does not change (an extension of Figure 3b).

Similarly the discussion focusses on the Mo-edges, but the shorter S-terminated edges appear to show extensive changes and variation in the different images. For example in Figure 5 there appears extensive variation in functionalisation, presence of vacancies, intensity variation (suggesting mono-S rather than S₂?) etc. The authors would need to also exclude these more quantitatively from their study to show

thiophene treatment is indeed functionalising purely the Mo-edges.

Other candidates are S₂ dimers and extrinsic impurities. There is often varying contrast along the image edges in the author's STM images, can the authors exclude S₂ dimers and also extrinsic impurities? (for example alongside the top and right corners of Figure 1a, but not the left). Although formally charged, S₂ structures could be stabilised by charge transfer from the gold substrate.

Secondly, the authors discuss variation in vacancy concentrations but do not provide sufficient statistics. How many flakes and sites were examined? And what was the statistical breakdown of particle size (side-length)? What fraction of examples have multiple vacancies on the same side? There is a brief discussion of coupled vacancy formation energy right near the end but this would be useful also when vacancy formation is first discussed. Detailed quantitative information on the statistics (including ideally expansion of the various Figure 2 images) is important, not least because it may be certain side-lengths (/particle sizes) dominate the catalytic behaviour. This is information which needs to be in the paper.

Figure 2d should be extended to also include the equivalent lines for A and M sites.

Concerning the theoretical calculations, these seem strong and I particularly appreciated Table S1. A few points:

- Could the authors quote the vacancy formation energy for periodic 1H-MoS₂ calculated this way, as a reference value? (and/or for continuous stripe models, as the authors already have these).
- H₂S is used as the reference source for sulfur in the formation energy calculation. Can the authors explain why (and not, for example, S₆ rings?)
- The Gaussian smearing used is quite high and suggests there were some issues achieving convergence (which could be linked to the spin). The authors say the formation energies were not significantly altered by spin but I suspect the energy difference would be higher with a much lower Gaussian smearing. Since the energies are not really being used quantitatively here it doesn't seem critical to the arguments in the paper.
- If possible, calculated STM images of the structures would improve the paper (there are freeware packages given the authors are using VASP).

Minor points:

There appears a difference between the modelling and experiment for the S-edge; it appears as if the central S-pair extend further out from the edge in the DFT image, whereas in Figure 1a the experimental image appears to show the inverse. Can the authors explain this.

Line 97 refers to Figure 1a but this does not really show the atomic structures as suggested in the text.

In general I find that it is a strong and well-constructed paper, but at the moment it is too focussed and lacking the "broad interest" aspect for a Nature Communications (but would be an excellent article in a

more specialised journal). If the authors were to convincingly address all of the wider points above, it might then be suitable for Nature Communications.

Reviewer #1 (Remarks to the Author):

In this paper, the authors investigate a very important catalytic reaction, hydrodesulfurization. Although a lot has been found about this process, still many details remain unclear. This paper sheds new light on HDS of thiophene, and especially on the question how it adsorbs to the catalyst to have its S atom abstracted. The insight that S atoms of the catalyst move to create space for the bulky thiophene is very interesting for the community. Therefore, I advise this paper to be published, after the following minor issues have been solved.

1. line 67: what are exact MoS₂ structures? Please rephrase.

Reply: Thank you for this suggestion. The intention with the sentence was to mention that we compare STM data with DFT calculations on particles with the same structure, i.e. considering edges and corners. Much of the previous work in the field has used DFT calculations of semi-infinite stripe models of MoS₂ which have limited the ability to compare with experiments, particularly in view of relaxation effects that only occur when the corner is explicitly included. We have now corrected the sentence to make our point clear.

Action: Rephrased as: “Herein, we use a combination of atom-resolved scanning tunneling microscopy (STM) and density functional theory (DFT) on corresponding MoS₂ nanoparticle structures to determine the precise adsorption configuration of thiophene on all types of CUS sites present on the MoS₂ edges and corners present under HDS conditions”

2. Fig. 1a: It is clear from the STM image that the number of vacancies is different on the three Mo edges. Is this a general observation for all MoS₂ nanoparticles? Has this been analyzed for many nanoparticles and can the difference be explained by statistical fluctuations? Or is there possibly another explanation? Please clarify this in the text.

Reply: The reviewer is correct that three vacancies occurring on one edge is an unusual observation, however, we note that it is not an unrealistic scenario. To substantiate this point, we can compare the expected number of vacancies on a Mo-7S particle from the values in Figure 2a. An Mo-7S particle (like in Figure 1a) has 6 C, 6 A and 9 M positions, respectively. This would on average lead to ~1.7 vacancies on M, ~1.5 vacancies on A and ~0.3 vacancies on C. Thus while the observation of 3A, 1M, and 0C vacancies shown in Figure 1a is higher than the expected average for A sites, it is still within reason. We further expect that the formation of a vacancy on one A site is uncorrelated with the presence of an A vacancy on the other end of a 7 S atom long edge, and as such the observation of two A vacancies on one edge is quite realistic. It is possible that formation of A vacancies is influenced by M vacancies on a next-neighbor site on the same edge (as in Fig 1a), but since observations with two or more vacancies on the same edge are rare, it is not possible for us to quantify such an effect at present. On the other hand, two neighboring vacancies (i.e. A and M) are distinctly disfavored (see theory part) and we also never observed that experimentally. Finally, we prefer to keep the presented image as it nicely illustrates the different types of edge vacancies.

3. Fig. 1c: in the bottom structure, the vacancies are named with M, but they are also adjacent to a corner, so A. Why do you choose for M, and not A? Is it because they behave more as M? Please clarify.

Reply: We thank the reviewer for remarking this. We agree with the reviewer that the middle sulfur vacancies in the smallest Mo-4S Mo-edges should be assigned as A. That would be more correct in all cases. We corrected the assignment of the Mo-4S edge to C-A-A-C, and went over the data again. The vacancy probability on A sites on the Mo-4S particles is 0.25, so this value does not change much, and the average vacancy probability on the M site is the same 0.18. These changes do not give rise to changes in our overall interpretation.

Action: We have updated the ball model Figure 1c. The statistical data in Figure 2a and Figure 3b have been updated to reflect the assignment of the Mo-4S edge to C-A-A-C, and the relevant numbers discussed in the text have been changed to reflect the slightly modified values from the re-counting.

4. Fig. 2d: label x-axis. For readers less familiar with HDS and its conventions it would be more clear to call it the Mo edge instead of M-edge.

Reply: We agree.

Action: This is now changed in Fig 2d.

5. Fig. 3a: add labels to the edges in the STM image.

Reply: We agree.

Action: The labels have been added to the image.

6. around line 221: if the adsorption of thiophene to certain sites is hindered by its structure, i.e. is a steric effect, how can it then be explained that the adsorption changes at higher temperature? Does the size of the vacancy change at higher T? The size of the thiophene will not change. Please clarify.

Reply: The temperature dependence in Fig. 3b indicates that an energy barrier is present for the adsorption of thiophene for some sites. The reviewer is correct that a strong temperature dependence is not expected from direct adsorption onto a vacancy as long as there is unhindered access, i.e. it would be hard to imagine that a vacancy becomes bigger and therefore more accessible with temperature. However, it is a key point of the paper that the favorable adsorption modes of thiophene occurs with a concurrent movement of neighboring S atoms to make space, which is indeed the case for adsorption on A and M sites. Such a movement will be associated with an energy barrier (for example the formation of an S2 dimer on a neighboring site), which thus explains the temperature dependence observed here. Adsorption on C sites is comparatively facile, and correspondingly we actually see that such sites seem to be filled already at room temperature in Fig 2b. Since this is an important conclusion, we have now made this point more clear in the manuscript.

Action: We have elaborated on this point as follows: “The onset of adsorption on A and M sites at higher temperature, compared with C, reflects a higher intrinsic kinetic barrier for thiophene adsorption on these sites. An energy barrier adsorption is expected for adsorption when the steric access is limited and activation may thus be required when relaxation at the adsorption site is part of the adsorption mechanism”

This is linked to the more specific discussion later in the modelling part where we provide a calculation of the barrier on page 16: “As shown in the supporting information (Supplementary Table S3), the barrier for S-dimerization starting from an edge vacancy and creating two adjacent vacancies without thiophene is very high (~2.1 eV). The barrier for the concerted step is 0.8 eV lower than the sequential pathway (at ~1.3 eV), which is fully consistent with the STM observation that thiophene adsorption on A and M sites is favorable, but with a significant reaction barrier”

7. line 231: how do you know that this is due to hydrogenolysis and not just desorption of thiophene because of the higher T?

Reply: The reviewer is right that direct desorption of thiophene would regenerate the same amount and distribution of vacancies. Such a process would be induced by increasing the temperature (and not by dosing hydrogen simultaneously). However, in our case, the experiment in Fig. 3b indicated with blue bars was already conducted by dosing thiophene at a high temperature (650K), and in this case we still see thiophene adsorbed. We thus conclude that elevated temperature does not lead to desorption, rather it facilitates adsorption. In comparison, heating to almost the same temperature (673K) and dosing hydrogen leads to many more vacancies, which makes it more likely that a process involving reaction with hydrogen is taking place. We agree that a full demonstration of hydrogenolysis would be interesting, but it requires more experiments (such as mass spectroscopy of the desorbing species) which was not possible here. We, however, feel that the discussion of hydrogenolysis in the paragraph is still relevant, since the hydrogenolysis and regeneration of vacancies is a key step in HDS.

Action: The sentence is now: “Since high temperature alone is not enough to desorb the thiophene (Fig. 3b, blue bars), we consider that the vacancy regeneration process is likely due to a hydrogenolysis reaction that produces an organic product and H₂S, thereby completing a full catalytic cycle.”

7. Fig. 4: please label the edges.

Action: Labels are added.

8. line 275: how does the thiophene adsorb over both the A and M sites? Probably not via the S atom to the vacancy. Please explain.

Reply: Here we would like to refer to the modelling results, in particular figure 4(iii) where the adsorption mode referred to in the discussion is displayed. The modelled adsorption configuration involves a double vacancy (generated by the displacement of another), so that S in the thiophene is located near one Mo and the carbon ring adsorb on the other one. This is the general case for all our modelling configurations where an S is displaced away to make room for the thiophene.

Action: We added this sentence: Note in figure 4(iii) that the S in thiophene coordinates to one of the undercoordinated Mo atoms whereas that carbon ring system is placed over the Mo which has become accessible due to the displacement of the S.

Fig. 5: this figure can be improved in the following way:

- a) remove more black so that MoS₂ can be bigger
- b) label S2 is unreadable

- c) mention in the caption what the blue arrow in a corresponds to
- d) mention in the caption the tunneling parameters

Reply: *We agree.*

Action: *We have included a modified version of figure 5 and the caption according to the suggestions of the reviewer.*

- 9. line 335: it is not clear to me how the authors conclude on the amount of V_s under reaction conditions since their work is performed under UHV conditions. If, however, this is my mistake, and their work has been performed under reaction conditions, this should be made clearer from the text.

Reply: *We have now made it clear that this conclusion is a result obtained from the theoretical modelling, and not the experiment. We refer to figure 3c.*

Action: *The revised sentence is now included as : “Our findings indicate that corner sites in a MoS₂ nanoparticle are very favorable for thiophene adsorption, but our modelling also shows that vacancy formation may be so energetically expensive that a very low fraction of the corner (C) sites will contain a V_s under steady state reaction conditions”*

Reviewer #2 (Remarks to the Author):

Salazar et al. report a model study of the industrially-relevant hydrodesulfurization on MoS₂ catalysts. A model catalyst, single-layer MoS₂ nano-islands grown on Au(111), and a model reaction, thiophene (hydro)desulfurization, were employed and studied by experimental (mainly STM) and theoretical (DFT) methods.

Distinct differences in the thiophene-catalysts interactions at preformed S-vacancies at edge and corner sites (and even differently within edges) were observed. Thiophene adsorption alone leads to restructuring of the S atoms on neighboring sites around the initial vacancy sites.

- 1. One could question whether the model compound thiophene really mimics the much more stable sulfur-containing compounds in gas oil feeds, but that's true for all model studies.

Reply: *The referee is right that thiophene is a relatively simple model of S-containing compounds in real HDS feed. Thiophene, however, is often preferred in reactivity modelling of HDS catalysts because it is a quite stable molecule with two C-S bonds and an aromatic nature that increases its stability. In terms of direct desulfurization, thiophene is expected to be a good model compound, and we expect that our findings are also valid for the reaction of dibenzothiophene, which still reacts predominantly by direct desulfurization. For even larger molecules with a sterically shielded S, such as 4,6-dimethyl-dibenzothiophene, it is an open question if an adsorbing molecule can generate its own larger adsorption site, as seen here. This is something which we would like to pursue in future studies.*

- 2. The work was performed on the highest technical level and all analysis and interpretations are thorough and sound. My main concern is rather that the very same

model system was studied for about two decades and the major breakthroughs of this model system have already been published in general high impact journals, whereas the further progress – although important – seems rather incremental and only relevant for specialist in the field.

Reply: We thank the reviewer of this positive assessment of the technical quality of our work. We agree that some aspects of this chemistry have been published in other journals, however, we note that the size and location dependent activity and the experimental demonstration of the dynamic rearrangements of the surface sulfur atoms have not previously been reported before. This level of detail is rarely studied in the general catalysis literature. We, therefore, believe that the results of this paper will gather the interest of a very broad audience.

Reviewer #3 (Remarks to the Author):

The authors present a combined STM-DFT study of thiophene absorption on edge sites of MoS₂ nanoparticles and suggest a mechanism of S-deplacement to allow incorporation of larger functional groups onto neighbouring edge sites.

In general I find that the subject of the paper is interesting, and the approach of combining STM with DFT studies is useful. However a problem with such an approach is that the images can be easily interpreted many different ways, and attention focussed just on specific aspects of the images. That appears to be the case with the current paper where the argument is heavily "driven" in favour of the model proposed by the authors, and alternative explanations and mechanisms are not covered and quantitatively dismissed sufficiently.

1. As some specific examples, the authors state there is no change in basal plane properties after exposure to thiophene, but do not provide any quantitative support for this statement. Basal plane vacancies appear visible in their STM images, and to justify their claim it would necessary to show for example statistics of basal plane vacancy concentration before and after thiophene treatment to demonstrate this does not change (an extension of Figure 3b).

Reply: We thank the reviewer for bringing this point up. We did not include basal plane vacancies in the statistical material, since we do not have indication that basal plane vacancies are induced by the hydrogen treatment and temperatures used here. We are aware that electrochemical conditions corresponding to hydrogen evolution (JACS (2016), 138: 16632 and more) and induced strain of MoS₂ monolayers may facilitate the formation of basal plane sulfur vacancies (e.g. Nature Materials (2016), 15: 1). Heating WS₂ monolayers at very high temperatures (>973K) may also induce S vacancies on the basal plane (Phys. Rev. Lett. (2019) 123, 076801). Recent reports (Nature Chemistry (2018), 10: 1246), and some of our own work (2D Materials (2019), 6: 045013), also shows O exchange on the basal plane in O atmospheres. However, these conditions are all significantly different from the temperatures relevant for HDS (typically 673K) and for the experimental conditions applied here (673K and hydrogen

at 10^{-4} mbar), which explains why an appreciable amount of basal plane S vacancies is not produced in our experiment.

The referee is, however, right in noting defects on some basal plane positions. We occasionally see basal plane defects such as the one in Fig. 1a. This image is a low-bias STM image reflecting the S lattice of MoS₂ and the defect is imaged with a dark three-fold symmetric appearance located on the metal lattice. The position and the threefold symmetry of this defect is more consistent with a defect located on the Mo-position of MoS₂ (see e.g. Nanotechnology 27 (2016) 105702)), and it could thus be associated with a Mo vacancy or a foreign atom located on the Mo position. Another type of defect results in a slightly more diffuse increase in brightness, which we attributed to a defect in the Au underneath, to which STM imaging is sensitive (see e.g. Nano Lett. 2016, 16, 8, 5163–5168)). Nevertheless, the number of these defects is very low and unaffected by the hydrogen treatment and we therefore prefer not to include them in our statistical account. The types and nature of defects in monolayer MoS₂ is a very important and interesting topic in relation to 2D materials too, but we feel that an extensive analysis and account of the relatively few basal plane defects would be less important in relation to the findings on edge reactivity in our paper here.

Action: We have made clear that defects located on basal plane sites are present, but conclude that they do not reflect sulfur vacancies and are therefore not included in the statistical counting. On page 5 we have included the following note: “Atomic defects are sometimes present within the basal plane too (e.g. in Fig 1a), but these are not induced by the hydrogen treatment and their location suggest an impurity atom on the metal lattice of MoS₂ rather than a basal plane V_S.”

2. Similarly, the discussion focusses on the Mo-edges, but the shorter S-terminated edges appear to show extensive changes and variation in the different images. For example in Figure 5 there appears extensive variation in functionalisation, presence of vacancies, intensity variation (suggesting mono-S rather than S₂?) etc. The authors would need to also exclude these more quantitatively from their study to show thiophene treatment is indeed functionalising purely the Mo-edges.
3. Other candidates are S₂ dimers and extrinsic impurities. There is often varying contrast along the image edges in the author's STM images, can the authors exclude S₂ dimers and also extrinsic impurities? (for example alongside the top and right corners of Figure 1a, but not the left). Although formally charged, S₂ structures could be stabilised by charge transfer from the gold substrate.

Replies to points 2 and 3: We agree with the referee that it is important to report our observations for the S edges as well, and we have now included a discussion of the S edge in the revised version. The S edges do not show changes from hydrogen treatment to the subsequent dosing of thiophene, which points to them being unreactive. In reference 50 (Nature Communications (2018), 9: 2211)) the stability of the S atoms on the S edge is presented in terms of DFT and STM results, and it is found that the S edge of MoS₂ is much less likely to have a reduced S coverage compared with the Mo edge. In fact, a key point in that paper was that Co induces changes to the S edge, which may explain why Co is the best promoter in HDS catalysis (it activates the S edge).

The figure included below compares STM data for the S-edge for the experimental series after hydrogen dosing and subsequent thiophene dosing (added as supplementary figure S3). There is no systematic variation to suggest adsorption of thiophene directly on the S edge. The referee is right in observing a systematic intensity variation of the protrusions located on the S positions on the S edge. The line scans show a 0.1\AA modulation between neighboring protrusions in all cases. This seems to be a general feature before and after thiophene. Theory modelling of the 100% S coverage S edge (see reply to question 10) indeed predicts an alternating splitting and dimer formation as shown in the side-view ball model in figure 2b, which is a likely explanation for the experimental observation of a modulated height on the S edge. This effect on the 100% sulfur covered S edge has also been noted in several other DFT studies, and a more detailed clarification of the experimental edge structure could be a good topic for a future study. Here we prefer to focus on the reactive Mo edges.

Figure S3: Comparison of line scans performed on the S-edges of $r\text{-MoS}_2$ and for the subsequent thiophene exposure series at 300K. Note that an S-edge with a 3 S atom edge length is present in the first image whereas S edges with two S atoms are present in the three subsequent images. The $\sim 0.1\text{\AA}$ height modulation between neighboring protrusions on the S edge is associated with the splitting and formation of S₂ dimers on the S edge, as illustrated in figure 2b (side view). We note, however, that the experiment suggests a dimer located in the middle whereas the model in figure 2b shows the inverse. This may be due to slight energy variations, possibly from interaction with the Au substrate (see also reply to point 10).

Action: We have included Figure S3 in the supplementary information to support our observation that thiophene does not seem to adsorb here in our experiment. We have also added the following sentence in the main text:

Page 10 “The corresponding short S-edges did not show changes in the edge structure upon thiophene exposure (see Figure S3), which is indicative of thiophene not adsorbing on the 100%S coverage S edge”

4. Secondly, the authors discuss variation in vacancy concentrations but do not provide sufficient statistics. How many flakes and sites were examined? And what was the statistical breakdown of particle size (side-length)?

There is a brief discussion of coupled vacancy formation energy right near the end but this would be useful also when vacancy formation is first discussed. Detailed quantitative information on the statistics (including ideally expansion of the various Figure 2 images) is important, not least because it may be certain side-lengths (/particle sizes) dominate the catalytic behaviour. This is information which needs to be in the paper.

Reply: *We thank the reviewer for pointing this out. In order to obtain a statistically significant data set from the quite extensive STM experiments, we decided to make a compromise in our experimental analysis on the coverage of S vacancies, to include the four (Mo-4S), five (Mo-5S) and six (Mo-6S) S-atoms-long Mo-edges, which were the most representative edges for all the experiments carried in this study (see new Figure S1). For example, we have analyzed 151, 261 and 205 Mo-edges for the Mo-4S, Mo-5S and Mo-6S to generate the vacancy count in Figure 3b, respectively. It is indeed interesting to make a break-down of each edge type, and therefore we provide the detailed data for each edge length in Figure S2. In Figure S2, there are small qualitative variations between the observations for each edge type (4,5 and 6S Mo-edge), but considering the error bars, we consider it to be fully justified to collate data for the three edge lengths. Furthermore, the observation, that thiophene readily adsorbs on C, whereas A and M sites require thermal activation, is qualitatively the same (Figure S2) for the subsequent experiment related with thiophene adsorption. Thus, we find no experimental evidence that one particular edge length dominates the reactivity.*

To keep the discussion of this point as coherent and concise as possible, we prefer to show the combined data in Figure 2b in the main manuscript, but we now refer to a new figure S2 in the supplementary which shows the break-down analysis of each edge type.

Figure S1: Distribution of S-edges and Mo-edge lengths reflecting the ensemble of synthesized *r*-MoS₂ nanoparticles. Left: Length distribution for the Mo and S-edges of MoS₂ nanoparticles. Right: Average MoS₂ nanoparticle morphology based on the total area of Mo and S-edges. A Mo edge length is denoted by its number of S atoms, e.g. as Mo-5S for an edge with 5 S atoms.

Figure S2: Break-down of data for sulfur vacancy coverage for Mo-edges of length Mo-4S, Mo-5S and Mo-6S as a function of the S vacancy position at M, A and C for MoS₂ nanoparticles exposed to H₂ at 673 K, thiophene at 300 K, 400 K and 650 K and post-annealed in H₂ at 673 K.

Concerning our considerations on the statistical location of vacancies on the “same” edge, we refer to the reply to reviewer 1, question 2.

Actions: We have included into our Supplementary Material the two new figures S1 and S2 to explain the statistics for all the Mo-edges analysed in each experiment for the Mo-4S, Mo-5S and Mo-6S and the sulfur vacancy evolution as a function of the S vacancy

position at M, A and C for MoS₂ nanoparticles exposed to H₂ at 673 K, thiophene at 300 K, 400 K and 650 K and post-annealed in H₂ at 673 K.

We have included the following sentences to refer the reader to the new supplementary information:

Page 6: “These sites define the possible edge positions on the Mo-edges lengths most abundantly present in the synthesized MoS₂ nanoparticle ensemble (see Figure S1), corresponding to 4-6 S monomers on the edge as shown in the side-view ball models in Figure 1c. A breakdown of our statistical materials from the experimental images into each edge length (Figure S2) shows only a very slight variation of the vacancy numbers with the edge length, so for the experimental analysis we consider C, A and M sites on differently sized edges to be similar in terms their vacancy formation probability”

Figure 2 caption is changed:... **(b)** Statistical plot showing the variation of the V_S fraction for each edge site position (M, A and C) obtained by counting S vacancies in atom-resolved STM images, compiled from images with 4,5 and 6 S atom positions on the Mo edge. The data reflects a temperature series where thiophene is dosed at 300 K (yellow), 400 K (red) and 650 K (blue), respectively. The green bins show the restored V_S vacancy distribution after post-reacting the thiophene-exposed sample in H₂ at 673 K. A statistical break-down for each cluster size is provided in Figure S2.

5. Figure 2d should be extended to also include the equivalent lines for A and M sites.

Reply: We thank the reviewer for this suggestion. We have now modified Figure 2d per the suggestion (and reproduce it below). We note that the vacancy formation energies on the corner site drops with edge length as expected from experiment. The vacancy formation energy of the M site seems site-independent, which is consistent with STM observations. Both STM and DFT show that the vacancy formation on A sites is site dependent; however, whereas STM shows a quite insignificant variation in the vacancy coverage on 5S than 4S or 6S, the DFT suggests that the sulfur on the A site of Mo-5S is the most difficult to remove (E_s of 1.19 eV on 5S compared to 1.12 and 0.99 eV for 4S and 6S). While more analysis is required (and is outside the scope of this manuscript), we suspect this discrepancy arises from the nature of the DFT functional to trimerize the Mo atoms on the edge which leads to considerable heterogeneity even among A sites. Further, sulfur atoms located on the longer bridge (e.g. between two groups of trimers in 5S or a trimer and dimer in 4S) have a smaller vacancy formation energy. The size dependence is now discussed on Page 9 in the manuscript.

Actions: The modified Figure 2 is as shown.

Figure 2: Site dependent vacancy formation on MoS₂ edges. **(a)** Bar plot showing the observed frequency of finding a V_S on corner (C), adjacent (A) and middle (M) sites as determined from atom-resolved STM images. **(b)** Top and side view ball model of the r -MoS₂ nanoparticle (5 S monomer long Mo edge) used in the DFT model. The calculated V_S formation energies (E_S) for the M, A and C edge sites, respectively, are given in eV (positive values are endothermic, see supplementary information). **(c)** Theoretically predicted V_S fraction (log scale) at each site as a function of the ratio of H_2 and H_2S partial pressures at 673K. **(d)** Size-dependent variation of the V_S formation energy for particle models exposing 4 to 6 S monomers along the Mo edge (corresponding structures are shown in section S4 of the supplementary information).

Further, on page 9: “The DFT modelling also predicts that the formation energies for S vacancies on C sites are dependent on the edge length. Figure 2d shows that the sulfur vacancy formation energy (E_S) at the corner site (see also Figure S4) decreases from ~ 2 eV to 1.8 eV (see Figure S4) upon going from Mo-edge length of four to six. Hence, this dependency will contribute to an increase of the overall V_S fraction compared with the 5 S-atom Mo-edge, as shown in Figure 2c (open circles). The increasing trend for vacancy probability for the corner sites is also noted in the break-down of the experimental data for different edge lengths in Figure S2. For comparison, the calculated E_S values at the M site in Figure 2d (only present in 5S and 6S Mo-edges) shows that the vacancy coverage should be size independent, which is consistent with our STM observations (Figure S2). Furthermore, for the A sites, DFT predicts a variation where the S atom is hardest to remove on the 5S Mo-edge (1.19 eV for 5S compared to 1.12 and 0.99 for 4S and 6S respectively in Figure 2d). We note that the DFT functional tends to trimerize Mo atoms particularly on the 5S Mo-edge which makes it generally easier to remove the S atom on the longer-bridge (e.g. M site of 5S) than others. We posit that this effect is the reason for variation of E_S values with Mo-edge length, which is not reflecting the variation in the experiment.”

Concerning the theoretical calculations, these seem strong and I particularly appreciated Table S1. A few points:

6. Could the authors quote the vacancy formation energy for periodic 1H-MoS₂ calculated this way, as a reference value? (and/or for continuous stripe models, as the authors already have these).

Reply: We thank the reviewer for this suggestion. Our calculated value for vacancy formation on the 50% Mo-edge of a periodic model (comprising of 6 Mo atoms in each row) is 0.99 eV; this is slightly lower than the 1.05 eV that was calculated for the M vacancy for the Mo-5S nanoparticle and is consistent with the vacancy formation energy (0.93 eV for going from 50% S to 33% S) reported by Rosen et al. (JPCC 2018, 122, 15318).

Action: We have included the following sentence: "For reference, the corresponding calculation of E_S on a semi-infinite stripe model was 0.99 eV, which is slightly lower than that of the M position".

7. H₂S is used as the reference source for sulfur in the formation energy calculation. Can the authors explain why (and not, for example, S₆ rings?)

Reply: The reviewer is indeed correct in noting that other reference sources for sulfur could be used in principle. However, H₂S is an appropriate reference here because vacancies are indeed formed upon hydrogen treatment with the chemistry likely being that molecular hydrogen picks up an S atom from the edge to form molecular H₂S in a multi-step reaction cycle. Further, the vacancies are filled upon treatment with hydrogen sulfide where the chemistry is likely the reverse of that for vacancy formation. It is therefore practical to relate the chemical potential of S with the ratio of the partial pressures of hydrogen and hydrogen sulfide. We, however, note that calculating the true S potential under the conditions of our study is non-trivial; this is precisely why we report a range of values for H₂:H₂S in Figure 2c. Using a different reference (e.g. solid sulfur) will unfortunately not overcome this issue.

8. The Gaussian smearing used is quite high and suggests there were some issues achieving convergence (which could be linked to the spin). The authors say the formation energies were not significantly altered by spin but I suspect the energy difference would be higher with a much lower Gaussian smearing. Since the energies are not really being used quantitatively here it doesn't seem critical to the arguments in the paper.

Reply: We appreciate the reviewer's concern about the value of smearing in the gas phase cluster calculations with large supercells should use small values of the smearing. In our experience, 0.05 is sufficient; to check this, however, we tried different smearing values (0.02, 0.005) and found that the energies differed by only ~ 1 meV..

To address the question of the effect of spin, we begin by noting a few of our and previously reported observations. The magnetic states of an MoS₂ nanoparticle are located on the 100% S-covered sulfur edge. This is consistent with a more detailed

study on magnetic edge states of MoS₂ reported by Vojvodic et al. (PRB 009, 80, 125416). Indeed, these authors show that (for their periodic model) the vacancy formation energy for 50% S-covered Mo-edge is only minimally affected by spin (~ 0.02 eV/Mo atom) while it is even smaller for the S-edge. Finally, we note that we did not carry out a rigorous study of the effect of spin on the total and relative energies of each calculation reported in this study since we are already considering spin polarization in our calculations. We have accordingly amended our note in the SI about the influence of spin to restrict ourselves to the discussion of the vacancy formation energy alone, which is the focus of this work. How the magnetic edge states are affected by the size of the nanoparticle and their impact on vacancy formation energies, the total energies, and catalytic rate constants is worthy of an independent study.

Action: “DFT Calculations” section in methods now says “Spin polarization is included in all calculations: the effect of spin was found to be negligible (~ 0.01 eV on the vacancy formation energy).”

9. If possible, calculated STM images of the structures would improve the paper (there are freeware packages given the authors are using VASP).

Reply: We concur with the reviewer on the general utility of simulating STM images and comparing them with experiments, however, simulated STM images of unsupported MoS₂ nanoparticles (considered in DFT calculations) may not be entirely representative of the experimental STM results obtained on Au(111) support. Specifically, we note that MoS₂ in and of itself is semiconducting while the edges of a nanoparticle of MoS₂ are metallic. Therefore, the STM simulation of a free-standing MoS₂ will capture the edge features but the basal plane itself will be featureless. Indeed, this is captured in the figure shown below. The figure is an isosurface plot corresponding to partial charge density computed using bands with eigenvalue within -1 eV of the Fermi level (this essentially corresponds to a bias of -1 V in an STM study). At constant current mode, STM essentially identifies the charge density isosurface corresponding to a specific current and bias. Clearly the charge density for an unsupported MoS₂ nanoparticle is located on the periphery of the catalyst particle. In view of these considerations, therefore, we decided not to simulate the STM images of our nanoparticles.

Figure: An isosurface plot of partial charge density (only bands within -1 eV of Fermi level are included). The isosurface is at 0.001 e/bohr³. The isosurface is shown in blue while the Mo and S atoms are shown in cyan and yellow respectively. The nanoparticle shown is Mo-5S (i.e. Mo-edge has 5S atoms).

Minor points:

10. There appears a difference between the modelling and experiment for the S-edge; it appears as if the central S-pair extend further out from the edge in the DFT image, whereas in Figure 1a the experimental image appears to show the inverse. Can the authors explain this.

Reply: The referee is correct in this observation. While the alternating formation and splitting of S₂ dimers is frequently observed on both S edges in cluster and in semi-infinite MoS₂ models, we currently have no good explanation for the inverted phase of the experiment in relation to the calculation. The energies involved in the dimer formation are not high (< 0.1 eV) and could further be influenced by the Au substrate, so a small fluctuation could likely induce the inversion. The simulated STM image above indicates that the effect is not electronic, i.e. due to an increased local-density of states on the S₂ dimer. This is not likely to change by including the Au substrate, but a specific account including the role of the Au is needed to solve in detail if charge donation from the Au could influence the dimers. As we do not consider thiophene adsorption or vacancies on the S-edge here, this would be more suitable for a separate study.

11. Line 97 refers to Figure 1a but this does not really show the atomic structures as suggested in the text.

Reply: This is now corrected.

In general I find that it is a strong and well-constructed paper, but at the moment it is too focussed and lacking the "broad interest" aspect for a Nature Communications

(but would be an excellent article in a more specialised journal). If the authors were to convincingly address all of the wider points above, it might then be suitable for Nature Communications.

REVIEWERS' COMMENTS:

Reviewer #1 (Remarks to the Author):

The authors have satisfactorily dealt with my concerns and suggestions and those of the other reviewers. It remains an open question whether this contribution is suitable for the general audience of Nature Communications or whether it is more suitable for a specialist journal, but I am happy to leave this decision to the editor.

Reviewer #3 (Remarks to the Author):

The authors have addressed all points raised by the three referees in a detailed and well-argued way. I have no problem with publication now in its current form.